# Validation of a 3D Local-Scale Adaptive Solar Radiation Model by Using Pyranometer Measurements and a High-Resolution Digital Elevation Model

**DOI:** 10.3390/s24061823

**Published:** 2024-03-12

**Authors:** Eduardo Rodríguez, Judit García-Ferrero, María Sánchez-Aparicio, José M. Iglesias, Albert Oliver-Serra, M. Jesús Santos, Paula Andrés-Anaya, J. Manuel Cascón, Gustavo Montero García, Alejandro Medina, Susana Lagüela, M. Isabel Asensio, Rafael Montenegro Armas

**Affiliations:** 1Instituto Universitario de Sistemas Inteligentes y Aplicaciones Numéricas en Ingeniería (SIANI), Universidad de Las Palmas de Gran Canaria, 35017 Las Palmas de Gran Canaria, Spain; eduardo.rodriguez@ulpgc.es (E.R.); albert.oliver@ulpgc.es (A.O.-S.); gustavo.montero@ulpgc.es (G.M.G.); rafael.montenegro@ulpgc.es (R.M.A.); 2Instituto Universitario de Física Fundamental y Matematicas (IUFFyM), Universidad de Salamanca, 37008 Salamanca, Spain; jgferrero@usal.es (J.G.-F.); smjesus@usal.es (M.J.S.); casbar@usal.es (J.M.C.); amd385@usal.es (A.M.); 3Departamento de Física Aplicada, Universidad de Salamanca, 37008 Salamanca, Spain; 4Departamento de Ingeniería Cartográfica y del Terreno, Universidad de Salamanca, 37008 Salamanca, Spain; mar_sanchez1410@usal.es (M.S.-A.); paula_andana@usal.es (P.A.-A.); sulaguela@usal.es (S.L.); 5Departamento de Matemática Aplicada, Universidad de Salamanca, 37008 Salamanca, Spain; josem88@usal.es; 6Departamento de Economía e Historia Económica, Universidad de Salamanca, 37007 Salamanca, Spain

**Keywords:** solar radiation, complex orography, shadow calculation, LiDAR data, high-resolution DEM, adaptive mesh, pyranometer measurement

## Abstract

The result of the multidisciplinary collaboration of researchers from different areas of knowledge to validate a solar radiation model is presented. The MAPsol is a 3D local-scale adaptive solar radiation model that allows us to estimate direct, diffuse, and reflected irradiance for clear sky conditions. The model includes the adaptation of the mesh to complex orography and albedo, and considers the shadows cast by the terrain and buildings. The surface mesh generation is based on surface refinement, smoothing and parameterization techniques and allows the generation of high-quality adapted meshes with a reasonable number of elements. Another key aspect of the paper is the generation of a high-resolution digital elevation model (DEM). This high-resolution DEM is constructed from LiDAR data, and its resolution is two times more accurate than the publicly available DEMs. The validation process uses direct and global solar irradiance data obtained from pyranometers at the University of Salamanca located in an urban area affected by systematic shading from nearby buildings. This work provides an efficient protocol for studying solar resources, with particular emphasis on areas of complex orography and dense buildings where shadows can potentially make solar energy production facilities less efficient.

## 1. Introduction

Knowledge of local solar radiation is essential for designing optimal solar energy systems. Ideally, this knowledge should come from long-term measured data. However, the limited coverage of radiation measurement networks imposes the need to develop solar radiation models. Therefore, validation of these models against measurements is crucial [1].

Dealing with solar radiation and shadows is a common issue in Geographical Information Systems (GIS). This stems from a wide range of interests that go from visualization and rendering to many potential applications such as predicting the radiation at crucial environments like glaciers [2], the design of agroforestry exploitations [3] or, in the urban environment, project planning [4], the generation of solar radiation maps [5] essential for the deployment of concentrating solar power (CSP) plants [6], and for planning the optimal location and orientation of PV or solar thermal panels for distributed generation of electricity or heat for domestic applications on building roofs or facades [7,8,9,10,11,12,13,14,15,16]. The latter application demands reliable models for solar radiation and shadow casting at the local level that can deal with the complexities of urban architecture, which has a tremendous impact on blocking the solar radiation of nearby areas. Furthermore, this involves a high accuracy of the input so that it can guarantee to capture the mentioned architectural features. On top of all this, a contained computation cost is highly desirable.

The Sun’s radiation in the direct, diffuse and reflected forms, as well as the treatment of shadow casting, is an active matter of research for which several methodologies have been proposed. Li and Liu [10] employed the Yallop algorithm to calculate the solar position and evaluate several methods for calculating diffuse radiation, but they did not address the issue of shadows cast by nearby obstacles. In [15], Toledo et al. report on the determination of the optimal PV panel tilting angle accounting for direct and diffuse radiation calculated by using the Erbs’s correlation and the effect of shadows cast by surrounding buildings parameterized with their height and distance. In [7,8], Dorman et al. proposed an open-source tool devised for modeling and shadow casting of a 2.5D vector-based environment. The shadow calculation is based on trigonometric relations between the Sun’s rays, the ground and any intervening obstacles. In [16], Tripathy et al. report on a system for computing the shadows as the intersection of parallelograms that arise from the transformation of a polygon-defined 3D city into a cloud of points with the terrain, facades and horizontal surfaces. Stendardo and coworkers [12] employed separate digital surface/elevation models with different accuracy levels for the terrain and for a 3D city in a system powered by ray-tracing algorithms implemented through parallelized GPU computations in order to estimate the solar potential in Geneva. Liang et al. [13] proposes a GIS-integrated open source tool that includes the *r.sun* package for the estimation of clear-sky, diffuse and reflected solar radiation that is in turn coupled to a well-established OpenGL-based 3D-graphics engine to compute shadow casting. In [14], Toledo and coworkers employed the SOL algorithm coupled to the direct and diffuse solar radiation model from Perez [17]. The algorithm, implemented in MATLAB, employs the concepts of *hyperpoints* to calculate the height of shadows in facades in a rectangular grid that represents the 3D city. In [4], Kaynak et al. obtain the direct solar radiation with the elevation angle constant (EAC) method and use the local meteorological with the Anstrom–Prescott model to estimate the sunlight attenuation due to atmospheric conditions, while the finite element method, back-face detection and ray-tracing algorithms are used to make a precise calculation of cast shadows.

Regarding the accuracy level of the Digital Elevation Model (DEM), several works suggest that a finely detailed 3D representation of the geometric features is vital to accurately evaluate the shadows cast by nearby geometries. In [18], the authors used various GIS light projection models to analyze how the level of detail in a building’s 3D approximation affects the quality of its projected shadows. They found that finer details are relevant for local applications such as predicting solar panels’ potential. In [19], the solar yield was assessed based on the grid spacing of the point cloud, finding out that the error increases linearly with the grid spacing up to 4 m, which is generally larger for structured grids than for unstructured grids, and that considerably fine grids are necessary when considering roofs with complex shadowing artifacts.

In the context of GIS and the representation of architectural features in the urban environment, a common approach is the use of the so-called 2.5D polygons, which consist mainly of a simplified representation of buildings as a composition of extruded shapes. The previous studies suggest that the polygon-based 2.5D urban mapping strategies might not be able to hold the needed detail when buildings that feature more complex shapes come into play. However, some works have extended the 2.5D geometry concept through the adaptation of more complex architectural shapes by using LiDAR data to reproduce the building geometry by means of a composition of arbitrarily and vertically (walls) oriented polygons within an adaptive mesh [20]. The present work uses a similar approach; we generate a high-definition DEM that accurately captures the terrain and the architectural geometry. This DEM is created from LiDAR data with a scan spacing of 0.30 m. Computing the shadows with this high-definition DEM is too computationally expensive, so instead, we use a triangular mesh adapted to it.

In this paper, we introduce a toolbox that accurately calculates solar radiation while considering the shadow casting of complex orography and architectural features. Section 2 describes this toolbox: the MAPSol model, the construction of the high-definition DEM and the generation of the adapted triangular mesh. In Section 3, we explain the process for acquiring experimental data and how we validated our model. We use a case study where shadows originate from the intricate geometry of the tower of the Cathedral of Salamanca onto a nearby placed pyranometer. The study’s results confirm our model’s accuracy in accounting for the shadows that impact solar radiation.

## 2. Materials and Methods

This multidisciplinary work has required a broad methodology covering different areas of knowledge. It begins by recalling some basic concepts about solar irradiance that will allow a better understanding of the work. Then, the MAPSol 3D local-scale adaptive solar irradiance model for which the validation protocol described in this paper has been designed is presented. This model allows the detection of the shadows cast by complex orography, but this requires a high-resolution map of the study surface. Therefore, the methodology used to obtain high-resolution DEM is described. To detect the shadows, the model relies on a mesh of the surface defined by the high-resolution DEM, so the next step is to generate an adapted mesh in the most efficient way. Finally, the model is validated by means of actual solar irradiance measurements. For this reason, the corresponding procedure of experimental data acquisition is described in detail.

### 2.1. Solar Irradiance

Irradiance is defined as the incident energy per unit of time and surface area, i.e., it is a power density [21]. The solar irradiance received on a point of interest depends on three key factors. The first factor is the Earth’s position concerning the Sun, determined by latitude, longitude, day, and time. This factor determines the angle at which the sunbeam strikes a given spot. The second factor is the orography and buildings, which impact shadows and the orientation between the sunbeam and the point of interest. The third factor is the atmosphere, which attenuates solar irradiance due to gases, solid and liquid particles, and clouds.

Depending on the factors considered, we can obtain two types of solar irradiance: *real-sky irradiance* (when all factors are considered) and *clear-sky irradiance* (when clouds are excluded) [22]. On the other hand, due to the scattering effects generated by the atmosphere and the effect of the angle of incidence, the global irradiance can be expressed as the sum of two components: the *direct irradiance* coming from the solar disk, and the *diffuse irradiance*, due to the scattering effects in the atmosphere.

The definition of the main irradiances is presented below [23].

Diffuse Horizontal Irradiance, DHI: This is the solar irradiance collected on a horizontal surface from the atmospheric scattering of light, excluding circumsolar radiation.Direct Normal Irradiance, DNI: It is the component of solar irradiance collected on a surface perpendicular to the Sun’s rays. The horizontal diffuse component, DHI, is neglected here. On clear days, this component is much larger than the diffuse component, while on days with high cloud cover, it is practically zero. As it is measured over the Earth’s surface, its values depend highly on atmospheric conditions and the time of the year.Global Horizontal Irradiance, GHI: This is the sum of all irradiance components collected over a horizontal surface. This includes the direct and diffuse components, as well as the reflected components, which are generally neglected because of their low value. The GHI can be calculated from the following expression:
(1)GHI=DHI+DNI·sinα
where α is the solar altitude angle, i.e., the complementary of the zenith angle of the Sun, DHI is the horizontal diffuse component, and DNI is the normal component.Beam Horizontal Irradiance, BHI: It is the direct horizontal component of the irradiance, i.e., the direct irradiance on a plane perpendicular to the vertical of the site. It can be obtained as follows:
(2)BHI=GHI−DHI

### 2.2. The MAPSol Model

This section describes the solar irradiance model MAPSol developed at the University of Las Palmas de Gran Canaria [24,25,26]. MAPSol computes the clear-sky beam irradiance in every triangle of an adaptative mesh representing the orography surface. What sets MAPSol apart from other models is that it considers the orography shadows in the irradiance computation.

#### 2.2.1. Clear-Sky Beam Irradiance Model

In this section, the main ingredients of the clear-sky beam irradiance model implemented in MAPSol are presented; a more detailed description can be found in [22,24].

The BHI equation for a horizontal surface is
(3)$$BHI=\left(I_{0}\epsilon \right)\left(-0.8662T_{LK}m\delta_{R}\left(m\right)\right)\left(L_{f}\sin\alpha \right).$$
In this equation, each of the terms can be related to the three factors impacting solar irradiance.

The first term $\left(I_{0}\epsilon \right)$ refers to the Earth’s geometry. $I_{0}$ is the solar constant (the Sunbeam irradiance at the mean solar distance) [27], and ϵ is a correction factor due to Earth’s orbit eccentricity [22,24].

The second term $\left(-0.8662T_{LK}m\delta_{R}\left(m\right)\right)$ concerns the atmospheric attenuation. Specifically, *m* and $\delta_{R}\left(m\right)$ are related to the attenuation by gas constituents, and $T_{LK}$ is related to the attenuation by solid and liquid particles [22]. The parameter *m* is the relative optical air mass [28]; $T_{LK}$ is the air mass 2 Linke atmospheric turbidity factor, and δR(m) is the Rayleigh optical thickness at air mass *m*, and both are calculated according to the improved formula in [29].

Finally, the third term $\left(L_{f}\sin\alpha \right)$ is related to the shadows of both the orography and the buildings. Parameter α represents the solar altitude angle, while $L_{f}$ is the lighting factor that considers cast shadows. A Solar Position Algorithm computes the solar altitude angle; the present model uses the one developed at the *Plataforma Solar de Almería*: the PSA algorithm [30]. On the other hand, the light factor takes values between 0 and 1; $L_{f}=0$ means complete shadowing and $L_{f}=1$ total sun exposure. The following section describes the algorithm to compute this factor.

#### 2.2.2. Shadow Detection

The model computes the light factor using a triangular mesh of the terrain generated with the method shown in Section 2.4. Therefore, $L_{f}$ relates directly to finding which triangles in the mesh are shadowed. This algorithm is described in detail in [24,26,31].

In short, this geometrical problem can be solved by finding those triangles that, looking at the mesh from the Sun, have at least another triangle that covers the former. This is done using a new reference system $x^{\prime }$, $y^{\prime }$ and $z^{\prime }$, where $z^{\prime }$ is in the direction of the beam irradiance (see Figure 1). Then, the mesh is projected on the plane $x^{\prime }y^{\prime }$. This is the view of the mesh from the Sun that allows us to find out which triangles are overlapping, if any.

To transform the coordinates and align the direction of the $z^{\prime }$ axis with the beam irradiance, we need the solar azimuth θ and solar altitude α (shown in Figure 1). These angles are obtained from the PSA algorithm [30].

Examining intersected triangles at each time step and applying temporal refinement to each analyzed triangle, a shading level is allocated to shadowed triangles based on the count of warning points located within other triangles (refer to Figure 2). This allocation is the light factor $L_{f}$.

The drawback of this technique emerges when attempting to apply it to a sizable mesh, as the computational cost becomes high, making the process too slow for some purposes.

In response to this challenge, a technique was devised in [31]. This approach involves pre-filtering candidate triangles before examining the ultimate mesh overlap. The determination of whether a triangle is shadowed is contingent upon either self-shadows or shadows cast by other triangles.

Self-shadows occur when a surface casts shadows onto itself, for instance, when it is oriented away from the Sun. Determining this condition is straightforward by examining the incidence angle ($\delta_{exp}$), defined as the angle between the solar vector and the normal to a triangle surface. The criterion for self-shadows is met when the incidence angle exceeds π/2. So, a *Self Shadows Light Factor* ($L_{fss}$) is defined with values of
$$L_{fss}=\left\{\begin{array}{cc} 0 & |\delta_{exp}|>\left(\pi /2\right)\\ 1 & \left(\pi /2\right)\geq |\delta_{exp}|\geq 0 \end{array}\right.$$

As the analysis of cast shadows demands substantial computational effort, an initial step involves circumventing the examination of all triangles oriented away from the Sun (self-shadowed). An effective strategy involves filtering triangles based on the following:In the absence of self-shadowed triangles (those facing away from the Sun), the entire mesh is illuminated, and no shadows are present.Only triangles oriented away from the Sun are capable of casting shadows. These are referred to as potential 1 triangles [31].

The shading analysis is finalized by examining the shadow projection on a specific set of points known as warning points (WPs). These points are uniformly distributed within the triangle using the Rivara 4-T algorithm [32] and correspond to the geometric centers resulting from each subdivision of the triangle (see Figure 2).

A warning point is shaded when its projection onto the $x^{\prime }y^{\prime }$ plane falls within a triangle that can potentially cast a shadow on it, and its $z^{\prime }$ coordinate is smaller than that of its projection onto the said triangle (it is further from the sun than its projection on the triangle). So, the *cast shadows light factor*, $L_{fcs}$ is computed as follows:(4)$$L_{fcs}=\frac{n_{wp}-i}{n_{wp}}$$
where $i=0,1,\ldots ,n_{wp}$ is the number of shadowed warning points, and $n_{wp}$ is the total number of warning points. The ultimate light factor, $L_{f}$ is
(5)$$L_{f}=L_{fcs}\cdot L_{fss}$$

### 2.3. High-Resolution DEM

There is a wide variety of technologies that allow a massive capture of terrestrial information [33]. Among the different geospatial data coming from different sensors that allow the massive capture of information, LiDAR data stand out [34]. LiDAR data, obtained from laser scanners on board different devices, are highly accurate three-dimensional data. Aerial LiDAR data are the most widely used data for massive data acquisition in urban environments [35]. Thanks to the INSPIRE (Infrastructure for Spatial Information in Europe) initiative [36] based on the Directive 2007/2 [37], many countries within the European Union provide LiDAR data to users. The National Geographic Institute (IGN) in Spain offers highly accurate and updated LiDAR data of the entire Spanish territory, captured periodically under the National Plan for Territory Observation (Plan Nacional de Observación del Territorio—PNOT) [38]. Previous studies have demonstrated the applicability of these LiDAR data in urban environments in Spain [39,40,41,42].

Although the IGN allows the user to download the LiDAR point cloud through the Download Centre [43], it also offers several three-dimensional products that are derived from LiDAR data: (i) Digital Terrain Model (DTM), (ii) Digital Elevation Model (DEM), and (iii) Digital Slope Model (DSM). All of them have one feature in common: a spatial resolution between 2 and 25 m. Although a spatial resolution of meters may be sufficient for some studies, in certain cases, a resolution of less than one meter is necessary to obtain accurate results. For the purpose of this research study, a high-resolution DEM was generated using the available LiDAR data from IGN over the study area, described in Section 3.2. The LiDAR data used belong to the first coverage of the PNOT observation program and have the characteristics described in Table 1.

From this data, a high-resolution DEM is generated by applying the grid method of triangulation, called the TIN (Triangular Irregular Network), which converts the network of individual points in a continuous surface of triangular facets. The vertices of the triangles are the points measured by the LiDAR sensor, forming the point cloud. The procedure behind the generation of the DEM is the Delaunay triangulation, where the length of the edges and the angles for the triangles can be limited in order to generate smoother surfaces with fewer peaks, avoiding sliver triangles. Thus, according to the Delaunay triangulation, triangles are generated for groups of three points, ensuring that there is no point inside the circumsphere of any triangle. Once the triangulation is generated, the triangular faces are interpolated, using the values of elevation and slope in order to generate a smoother elevation layer.

### 2.4. Mesh Generation

The MAPSol model described in Section 2.2 requires a discretization of the orography provided by the high-resolution DEM described in Section 2.3. The uniform triangulation associated with the DEM-raster should be discarded, as it is not computationally feasible. As an alternative, the construction of an adapted mesh is proposed, which allows capturing the singularities of the study domain and computing the beam irradiance in a more efficient way.

The mesh generation is based on a local refinement driven by the error (vertical distance) between the approximation of the orography (adapted mesh) and the actual orography (DEM). The algorithm starts with a coarse mesh and locally refines [32] those triangles that do not satisfy the error criterion. It produces graded meshes that capture the features of the domain.

The process is fully automatic and only requires an initial discretization, in this case the DEM, to capture the orography with a fixed accuracy. It has been implemented with the Neptuno library developed by L. Ferragut [44]. The entire mesh generation process, including the DEM construction described in Section 2.3, is summarized in Figure 3.

Although MAPSol does not require a high-quality mesh, the previous procedure could be combined with surface parameterization and/or smoothing techniques [45,46] to improve the mesh quality and reduce the number of triangles of the final mesh.

### 2.5. Experimental Measurements of Solar Irradiance with Pyranometers

The purpose of this section is to obtain a realistic picture of solar radiation in the area of the *Trilingüe* building of the University of Salamanca (Spain) through experimental data. The physical principles of the pyranometers used to collect solar irradiance data are presented below.

The main function of pyranometers is to convert a flux of incident photons into an electrical signal through a process of collection, selection and detection [47]. Depending on the magnitude to be measured, they require different adaptations. Here, the focus is on devices for measuring GHI and DHI. In the case of GHI, the devices for measuring global radiation register very unequal fluxes, and they also have a strong dependence on the Sun position. In order for the electrical signal obtained to be representative of the incident flux, they have to mix the radiation inside, so that the resulting signal is isotropic. This can be achieved by using a diffuser sheet or an integrating sphere as receiving elements.

On the other hand, pyranometers dedicated to measuring DHI require a complement that eliminates the direct component of the radiation. The standard procedure for determining diffuse radiation is by using a pyranometer (see Figure 4 and paragraphs below) shadowed by a shadowing ring, made of an opaque material of a calibrated width that avoids direct radiation on the receiver. The ring is positioned so that its axis is parallel to the Earth’s axis in the equatorial coordinate system and passes through the receiver. Figure 4 schematizes the positioning of a shadow ring at a latitude of 40° N, where the arrows on the right side represent the Sun’s rays. Once the ring has been correctly positioned, the receiver will be permanently shaded provided that the ratio of the receiver’s distance from the center of the ring to its radius is close to the tangent of the Sun’s declination. Due to the changing declination of the Sun over the course of the year, the shadow ring requires small adjustments in its position every two or three days, aiming to keep the receiver permanently shaded.

To compensate for the defect in the diffuse irradiance collection generated by the shadow ring, a correction factor is introduced. It depends on the latitude of the observation point as well as the time of the year. In reference [48], the ring manufacturer provides the full derivation of the correction factor, *C*, as follows:(6)$$C=\frac{1}{1-S}$$
where *S* is obtained as
(7)$$S=\frac{2\cdot 0.185\;rad}{\Pi }\cos\delta \;\left(U_{0}\cdot \sin\lambda \cdot \sin\delta +\sin U_{0}\cdot \cos\lambda \cdot \cos\delta \right)$$
where λ is the angle of latitude, δ the angle of declination of the Sun, and $U_{0}$ is the angle formed by the Sun’s position at sunrise and noon in the plane of the ring.
(8)$$U_{0}=\arccos\left(-\tan\lambda \tan\delta \right)$$

There is a wide variety of pyranometers that implement different technologies depending on the type of transducer they use. They are subdivided into two classes: indirect conversion detectors and direct conversion detectors [47].

Indirect conversion detectors: They work by converting the incident photon flux into another type of flux (usually heat), but they can also be a secondary photon flux. Heat flux detectors are widely used and their operation is relatively simple. To convert the photon flux into heat flux, a highly absorbing paint or varnish is applied to the detector, which causes its temperature to rise when the light beam is impinging on it. Knowing the temperature at two points and assuming that the steady state is reached, the intensity of the flux is calculated, which will be proportional to the temperature difference. Figure 5a shows a general scheme of the parts of an indirect heat flux conversion pyranometer. In the upper part there are two domes, the outer dome has the function of avoiding energy exchanges due to convective phenomena; as a whole, the domes act as an integrating sphere. As can be seen, the detector is surrounded by an anti-radiation shield to prevent radiation penetrating from anywhere other than the dome. Figure 5b shows the Pyranometer Kipp and Zonen SMP10, belonging to the Energy Optimization, Thermodynamics and Statistical Physics Group (GTFE), with which the Global Horizontal Irradiance measurements were performed.Direct conversion detectors: Again, there are two types. Photoemitter cells are based on the junction of an anode and a cathode, between which there is a large potential difference (in the range of kV), and an avalanche effect is produced. On the other hand, there are detectors based on PN junctions, the photodiodes, where the current generated is proportional to the incident flux. These types of detectors have better sensitivity than avalanche detectors and work with low voltage [49].

**Figure 5 sensors-24-01823-f005:**
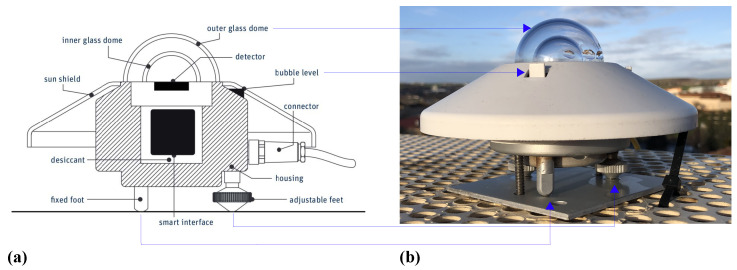
Heat flux sensing pyranometer: (**a**) basic scheme (taken from Kipp and Zonen. Instruction Manual SMP Series [50]) and (**b**) pyranometer belonging to the Group of Energy Optimization, Thermodynamics and Statistical Physics (GTFE) of the University of Salamanca.

For this work, radiation data obtained in situ are used. These data are recorded with two Kipp and Zonen pyranometers, from model SMP10. These are class A pyranometers (according to ISO 9060) that implement thermopiles as heat flux detectors. Thus, they work as indirect conversion detectors. With them, different magnitudes are collected: pyranometer 1 collects GHI measurements, while pyranometer 2, equipped with a Kipp and Zonen CM 121 shadow ring, takes DHI values. Table 2 shows the main characteristics of the model in question (detailed information about the pyranometers or the shadow ring can be found in the complete manufacturer’s manual [50]).

Figure 6 shows a comparison between the wavelength response of an SMP series pyranometer and the standard solar spectrum collected at sea level. As can be seen, the pyranometer response is practically linear over the entire range of interest, from the infrared (around 2000 nm) to the ultraviolet limit of the visible (around 350 nm). In this region of the spectrum, most of the energy from the Sun is deposited.

## 3. Results

In this section, the case study with which the MAPSol model has been validated is precisely described, including the description and reliability of the chosen irradiance data and the generation of the high-resolution DEM of the study area, as well as the adapted grid, and the simulation with the MAPSol model. The chosen case study provides the possibility to assess the effect of shadows caused by the intricate orography generated by buildings close to the location of pyranometers.

### 3.1. Experimental Data Acquisition

Experimental data were obtained from two pyranometers and a weather station, shown in Figure 7, installed on the rooftop of the *Trilingüe* building at the Faculty of Sciences (40.96062° N, 5.67075° W) of the University of Salamanca (Spain): a pyranometer Kipp and Zonen SMP10 (Figure 7a) and a pyranometer SMP10 with shadow ring model Kipp and Zonen CM 121 (Figure 7b).

During the day, the devices perform the corresponding measurements, storing GHI and DHI values every five minutes. At the end of the day, the collected data are sent to the computer, generating a *csv* file containing the irradiance information. The next step is to classify and filter the files. To do this, a program developed in “Mathematica^®^ Software v. 13.1, Licensed to Universidad de Salamanca”, identifies the files that belong to the same day and extracts the desired information from each of them. Finally, the program generates a new output file where all the information for the same day is merged.

The results presented in this work have been obtained with data between 30 July 2020 and 22 February 2023. In Figure 8, illustrative monthly averages have been performed on the irradiance, so that for each month, GHI and BHI curves are obtained as a function of time (which is represented in the UTC format). The monthly average is obtained by considering the data of all the days belonging to that month in the whole study period.

As previously mentioned, pyranometers take measurements of GHI and DHI. To obtain the DHI component, a correction factor which depends on the day of the year is required. For each of the measurements, the correction factor is calculated as a function of the solar declination angle, and hereafter, DHI is considered to include the correction factor. The BHI and DHI measurements are obtained, every 5 min, from Equations (Equation 1) and (Equation 2), respectively. During the night, GHI and DHI values should be identically zero, but it is found that negative values appear. This is usual when performing this type of measurements. However, it does not make physical sense to work with negative irradiances. To solve this problem, the following filters are introduced into the program:If $DHI_{i}<0\Rightarrow DHI_{i}=0$If $GHI_{i}<0\Rightarrow GHI_{i}=0$If $\left(GHI_{i}-DHI_{i}\right)<0\Rightarrow \left(GHI_{i}-DHI_{i}\right)=0$If $\sin\;\alpha_{i}<0\Rightarrow DNI_{i}=0$

The irradiance curves introduced below are intended to be as close as possible to the actual measurements, so no statistical treatment beyond averaging is carried out. Figure 8 shows the irradiance curves for four representative months of the year: March, June, September and December. In all these plots, a certain level of noise is observed within the irradiances, which evinces that the pyranometers are very sensitive to small changes. For example, on a summer day with isolated clouds, if a cloud comes between the Sun and the pyranometer for a few minutes, the direct components will become very small. When the cloud has passed, the direct component will again reach high values. When observing the set of figures, it can be noticed that the irradiance curves increase in width and height in the summer months. On the contrary, during winter, when the Sun is up for less time and at a lower height above the horizon, the curves become narrower and present lower peak values.

In order to assess the reliability of the measurements obtained, these values are compared with two independent external sources: AEMET [51] and Solargis [52] for solar irradiance. The Atlas of Solar Radiation in Spain, using data from the EUMETSAT climate FAS developed by AEMET [51], uses satellite data with a spatial resolution of 3×3 km. Data from the period (1983–2005) were used to elaborate the atlas. In order to compare the results in a more convenient way, Table 3 shows the results of this work compared with those of AEMET. The relative differences obtained for 9 of the 12 months of the year do not overcome 10% in the daily accumulated energy for the global component (GHI). This number is reduced to eight for the direct horizontal component (BHI).

Taking the experimental data of GHI and BHI from Table 3, it is possible to obtain the annual cumulative values and also the average value per day of the energy received. The annual cumulative values are easily obtained by performing the following operation:(9)$$E_{x}=\sum_{i=1}^{12}x_{i}N_{i}\;.$$
where *x* represents the irradiance component, the subscript *i* is the number of the month, and $N_{i}$ is the number of days in month *i*.

The Atlas of Solar Radiation in Spain [51] using data from the EUMETSAT climate FAS provides daily average GHI and BHI values in the form of an irradiance distribution map. The values extracted from these maps can be seen in Table 4, together with the values recorded in situ.

The value for GHI obtained in this work is within the AEMET interval, touching the upper limit. The experimental BHI component is 0.01 kWh $m^{-2}\cdot {\mathrm{day}}^{-1}$ above the maximum value of the interval. AEMET does not offer the annual average accumulated values, but they can be calculated by multiplying the daily average by the number of days in the year (the result is also shown in the same table). Again, the experimental value of GHI is within the range, while the value of BHI is slightly above it.

As it is known, a Sun chart is employed to present, at a specific location, the apparent position of the Sun, i.e., the height of the Sun at any hour of the day. From a Sun chart elaborated by the University of Oregon Program [53], together with a panoramic photo from the *Trilingüe* building at the University of Salamanca, Figure 9 is elaborated. This figure shows the Sun path chart from the *Trilingüe* building at the University of Salamanca (40.96062° N, 5.670759° W) between 21 December and 21 June, overlapped with a panoramic photo, taken from a pyranometer that registers GHI to identify shadowing sources.

One of these shadowing sources on the location of the pyranometers is the 93-m-high cathedral tower, located 296 m to the east. As can be seen in Figure 9, the cathedral tower casts its shadow in this area during the early hours of some days of the year, specifically between 22 August and 11 September, and in its symmetrical months, February and March. In these months close to spring, cloudiness is higher, so it is more difficult to obtain irradiance measurements that allow the effect of the cathedral’s shadow to be clearly seen. However, in summer, the sky is clearer, and this phenomenon can be seen more clearly in the measurements obtained.

### 3.2. Area Study, High-Resolution DEM and Adapted Mesh

An area of 430 m × 176 m has been selected which includes the *Trilingüe* building at the Faculty of Sciences of the University of Salamanca, where the pyranometers are located, and the cathedral tower located at 296 m in a straight line to the east, which casts its shadow on the pyranometers affecting the reading of the irradiance data.

Provided with the point cloud available in the area under study, the DEM generated with the process described in Section 2.3 presents a resolution of 33 cm (Figure 10). This methodology performs the generation of the Delaunay triangulation using negligible computing time (3 s for 35,228 points and 1,385,800 triangles), with a computer with Intel Core i7-6700 processor at 3.41 GHz, 64 bits, 32 Gb. RAM and 931 Gb (Dell Precision Tower 3620, Round Rock, TX, USA).

The size of the raster file is too large for the calculations involved in the MAPSol model, so the mesh is adapted using the procedure described in Section 2.4, achieving a reduction of 62% in the number of cells. The adapted mesh for a tolerance of 1 m has 529,503 triangles and includes all the singularities of the study area, in particular the cathedral tower. This element holds significant importance in the DEM. It is not present in some of the raster files, such as those provided by the IGN (Digital Surface Model–Building and Digital Surface Model), due to their low resolution and dimensions.

The number of triangles can be further reduced without affecting the accuracy of the shadow and irradiance calculations, using a tolerance of 5 m. This means that in the mesh adaptation process, a triangle is refined if the distance from the triangle to the high-resolution DEM is greater than 5 m, or 1 m in the previous case. The number of triangles of the adapted mesh with a tolerance of 5 m is 44,313, which represents a reduction of more than 95% with respect to the original mesh, without, as we will show further, a significant loss in precision in the calculation of shadows and irradiance, but with a considerably lower computational cost.

Figure 11c displays the fine adapted mesh (1 m) for the entire study area and several zooms of both meshes, the fine adapted mesh (Figure 11b) and the uniform original mesh (Figure 11a), for the Cathedral building area. A 3D reconstruction of the study area is depicted in Figure 12, including the location of the pyranometers with a red point, which allows the complexity of the study area to be observed. Furthermore, this procedure demonstrates that with few free resources, it is possible to reconstruct a 3D image of a very complex area such as this one, in the old part of the city of Salamanca. For Figure 12, the most appropriate orientation has been chosen so that the orthophoto of the area projected on the fine adapted mesh provides the best visual result.

### 3.3. Simulation with MAPSol

Using the coarse adapted mesh (5 m), the shadows and global irradiance in each triangle have been calculated for each day for a full year, with a time step of 5 min, with the MAPSol model. Computations have been performed in a computer equipped with two AMD EPYC 7313 CPUs, 128 GB of RAM memory and Debian Linux version 11 operating system. MAPSol is written in Python and C++, using the latter for the core of the computation and the former mostly to deal with input/output. In this particular work, Python 3.9.16 version and GNU g++ 10.2.1 C++ compiler have been used. Simulations have been performed in parallel, using one core per month. The mean wall clock execution time was 3 h and 56 min, including writing results to files for later analysis.

For each time step, a VTK file representing irradiance in the whole domain is written, as well as a csv file with values of GHI in the location of the pyranometers. Monthly average values of global irradiance in the study area have been calculated, and in Figure 13, four of the most significant months of the different seasons of the year have been represented. In addition, the annual mean GHI is depicted in a 3D image in Figure 14; notice that a view from the south has been chosen so that the facades most exposed to solar radiation can be appreciated.

Optionally, additional VTK files representing shadows cast in the domain for each time step can be obtained. Figure 15 represents the computed shadows cast in the study area on 4 September 2022 at 7.00 a.m., when the shadow of the cathedral tower, located in the blue dotted square, affects the reading of the pyranometers, located in the red dot.

### 3.4. Comparison of Simulation Results with Experimental Data

Taking into account the availability of data, due to cloud cover, possible technical difficulties and the reduced number of days on which the effect of the cathedral tower on the location of the pyranometers can be seen, the following dates have been chosen to compare the measured and calculated GHI: 4 and 11 September 2022, when the measurements will show the effect of the shadow of the cathedral, and 15 March 2021 and 4 August 2022, when the shadow phenomenon will not occur.

Both measured and calculated GHI values have been taken every 5 min throughout the 24 h of the day, with a total of 288 data points. In the two lower graphics of Figure 16, it can be clearly appreciated that the model captures the effect of the shadow of the cathedral in the early morning. The cathedral tower shadows the pyranometers just at sunrise, causing an interruption in the increase in irradiance, either with a sharp and short decrease (Figure 16, bottom left) or with a sharp and delayed sunrise (Figure 16, bottom right). Both phenomenologies are well captured in the simulation, with the pyranometer measurement (purple line) matching the calculated GHI (green line) very accurately. The upper graphics correspond to two dates where the shadow of the cathedral does not affect the irradiance data reading. The upper left graph shows the effect of cloud cover around 9 a.m. on pyranometer measurements (purple line).

To compare the accuracy of the results and the computational cost, the selected days were also simulated using the high resolution (1 m) mesh. The results at the pyranometer’s locations are the same (with 6-digit accuracy) as those obtained using the 5 m mesh, but the mean execution time is 68 times higher, so the 5 m mesh was chosen for all irradiance simulations. However, the use of the high resolution mesh might be appropriate in case a very precise study of shadows over the whole domain is intended.

In order to evaluate the precision of the model for a clear sky, four representative statistical error indicators have been calculated for the selected dates.

MAE: Mean Absolute Error
(10)$$MAE=\frac{\sum_{i=1}^{n}|{\hat{GHI}}_{i}-GHI_{i}|}{n}$$NMAE: Normalized Mean Absolute Error
(11)$$NMAE=\frac{\sum_{i=1}^{n}|{\hat{GHI}}_{i}-GHI_{i}|}{n\times GHI_{max}}\cdot 100$$RMSE: Root-Mean-Square Error
(12)$$RMSE=\sqrt{\frac{\sum_{i=1}^{n}{\left({\hat{GHI}}_{i}-GHI_{i}\right)}^{2}}{n}}$$NRMSE: Normalized Root-Mean-Square Error
(13)$$NRMSE=\sqrt{\frac{\sum_{i=1}^{n}{\left({\hat{GHI}}_{i}-GHI_{i}\right)}^{2}}{n\times GHI_{max}^{2}}}$$$R^{2}$: Coefficient of determination
(14)$$R^{2}=1-\frac{\sum_{i=1}^{n}{\left({\hat{GHI}}_{i}-GHI_{i}\right)}^{2}}{\sum_{i=1}^{n}{\left({\hat{GHI}}_{i}-\bar{GHI}\right)}^{2}}$$

Here, $GHI_{i}$ and ${\hat{GHI}}_{i}$ represent, respectively, the measured and calculated value of the global irradiance at time $t_{i}$, corresponding to the n=288 time instants for which data are available. $GHI_{max}$ refers to the maximum measured GHI, and $\bar{GHI}$ is the mean of the measured GHI. Table 5 summarizes the statistical indicators described above for the four selected days, showing very small values, especially the normalized errors, resulting in a higher coefficient of determination ($R^{2}\geq 0.99$).

## 4. Discussion and Conclusions

The combination of different technologies from diverse areas of knowledge has allowed the validation of a solar radiation model that is particularly useful in complex orographies such as those with buildings in an urban area. This is especially useful for the optimized design of solar installations that are increasingly being installed in cities.

The solar radiation model MAPSol allows us to estimate the clear-sky beam irradiance and generate a very detailed shadow map, which is computed at a reasonable computational cost by pre-filtering the triangles of the surface mesh and using an adapted mesh.

For areas with complicated orography, a high-resolution DEM is required, which is calculated from LiDAR data with a resolution of 33 cm. Adapting the mesh associated with this DEM allows the number of elements to be reduced by more than 90% without reducing the accuracy of the calculations, but at a much lower computational cost. The comparison of results and computational cost using two adapted meshes of different resolutions, one finer (1 m tolerance) and the other coarser (5 m tolerance), allows us to conclude that using the coarse mesh considerably reduces the computational cost without losing accuracy.

The accuracy of the MAPSol model has been validated by comparing the calculated GHI data with data measured by two pyranometers. The calculated GHI accuracy is close to 99% on days with low cloud cover. It is worth noting that on some of these days, the sensors detect the shadow of the dome of the cathedral tower in the early hours of the day.

Therefore, the proposed procedure and model allow the calculation of irradiance in areas of very complex orography with a reduced computational cost and freely available data, achieving a great balance between efficiency and effectiveness.

In addition, the adapted mesh generated from the high-resolution DEM has proven to be useful for recreating 3D images of complex areas with limited resources and freely available data.

In the future, the described procedure and model will be used to elaborate a solar map of a larger area of the city of Salamanca. In addition, the comparison of the measured irradiance values, subject to variable cloud cover, and those calculated for clear skies, can be used to produce cloud cover indices, and progress in the use of the MAPSol model under real sky conditions.

## Figures and Tables

**Figure 1 sensors-24-01823-f001:**
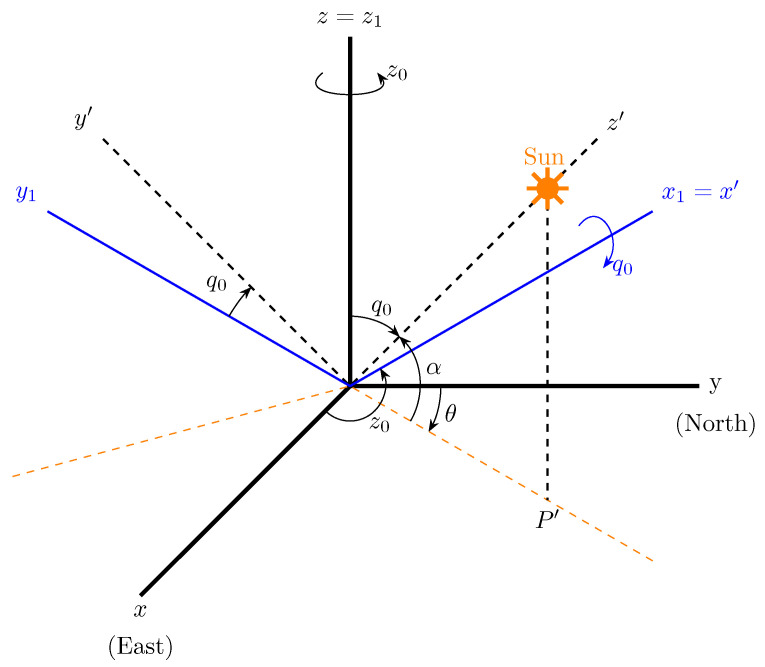
Reference system and Euler angles.

**Figure 2 sensors-24-01823-f002:**
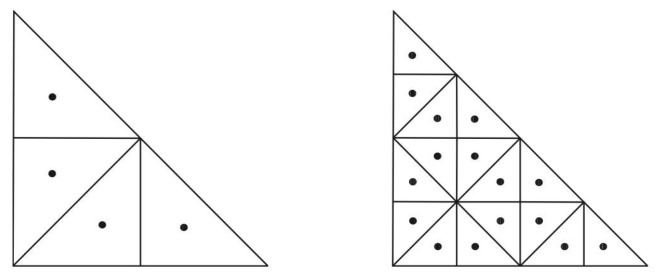
Warning points for shading are equidistributed over each triangle (**left** graph) simply using the centers of the triangles obtained by refining with the 4-T Rivara algorithm (**right** graph) [32].

**Figure 3 sensors-24-01823-f003:**
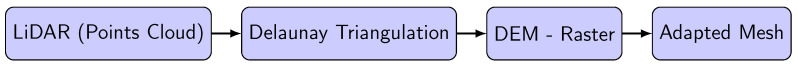
Scheme of the mesh generation procedure.

**Figure 4 sensors-24-01823-f004:**
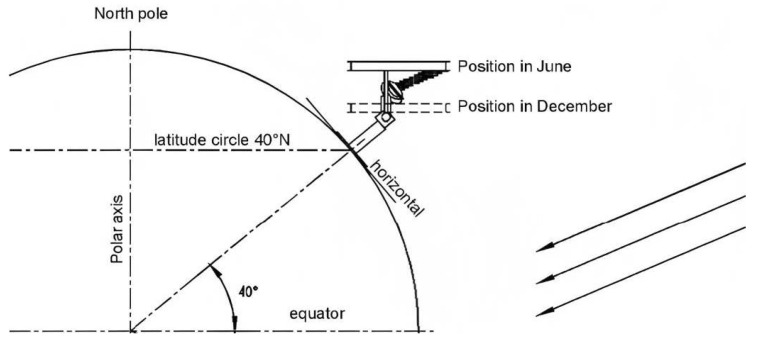
Schematic of shadow-ring (positioning, taken from Kipp and Zonen. Instructions Manual CM121 Shadow Ring [48]).

**Figure 6 sensors-24-01823-f006:**
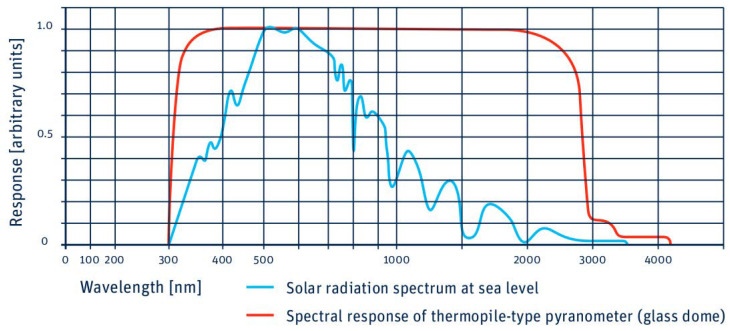
Spectral range response of a pyranometer of the SMP series manufactured by Kipp and Zonen versus the radiation spectrum at sea level (taken from Kipp and Zonen. Instruction Manual SMP Series [50]).

**Figure 7 sensors-24-01823-f007:**
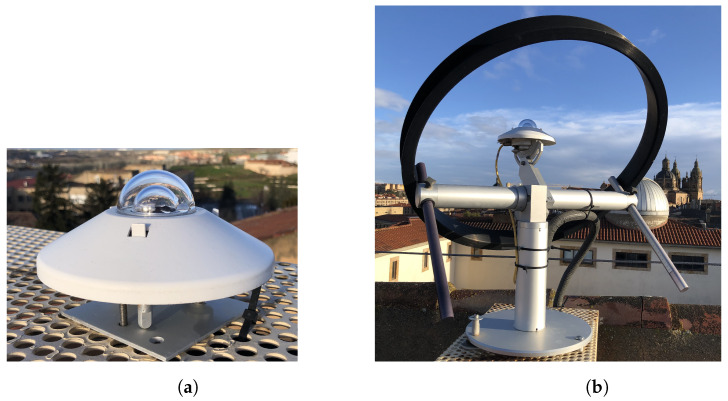
Measurement devices installed on the rooftop of the *Trilingüe* building of the Faculty of Sciences (40.96062 N, 5.67075 W) of the University of Salamanca (Spain): (**a**) pyranometer Kipp and Zonen SMP10 and (**b**) pyranometer SMP10 with shadow ring model Kipp and Zonen CM 121.

**Figure 8 sensors-24-01823-f008:**
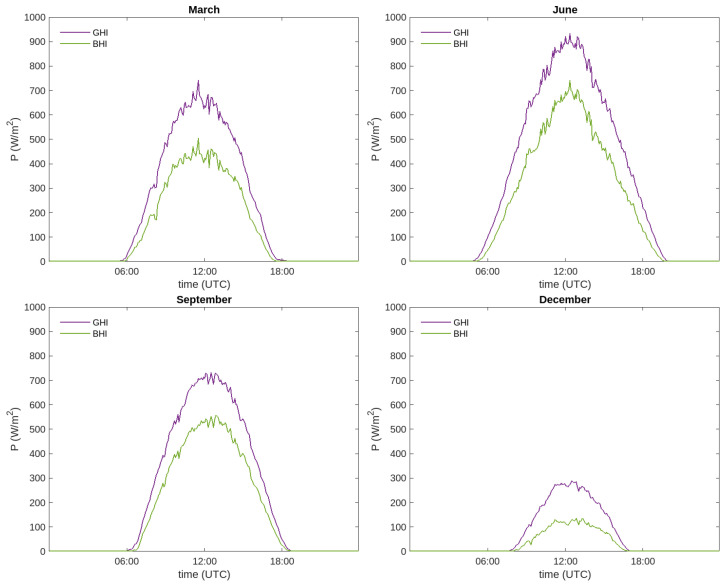
Averaged curves of GHI and BHI for different representative months of a year.

**Figure 9 sensors-24-01823-f009:**
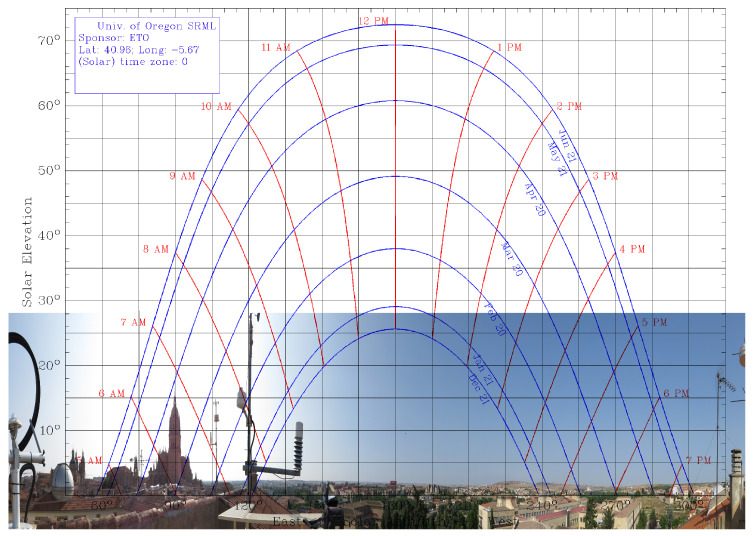
Sun path chart: Apparent position of the Sun from the *Trilingüe* building at the University of Salamanca (40.96062° N, 5.670759° W) between 21 December and 21 June [53]. A panoramic photo, taken from pyranometer that register GHI, is overlapped aiming to identify shadowing sources.

**Figure 10 sensors-24-01823-f010:**
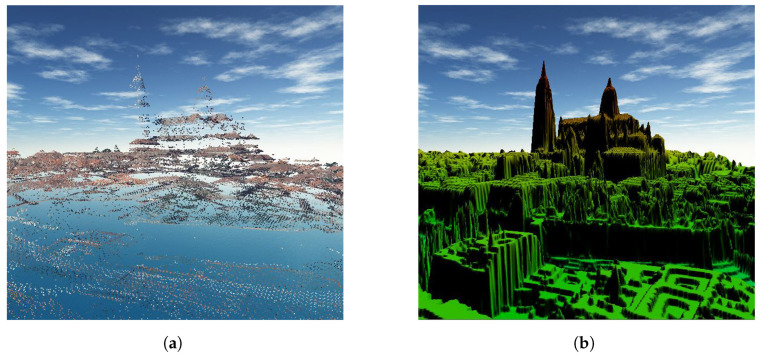
Three-dimensional view of the study area: (**a**) original point cloud and (**b**) DEM derived from the point cloud.

**Figure 11 sensors-24-01823-f011:**
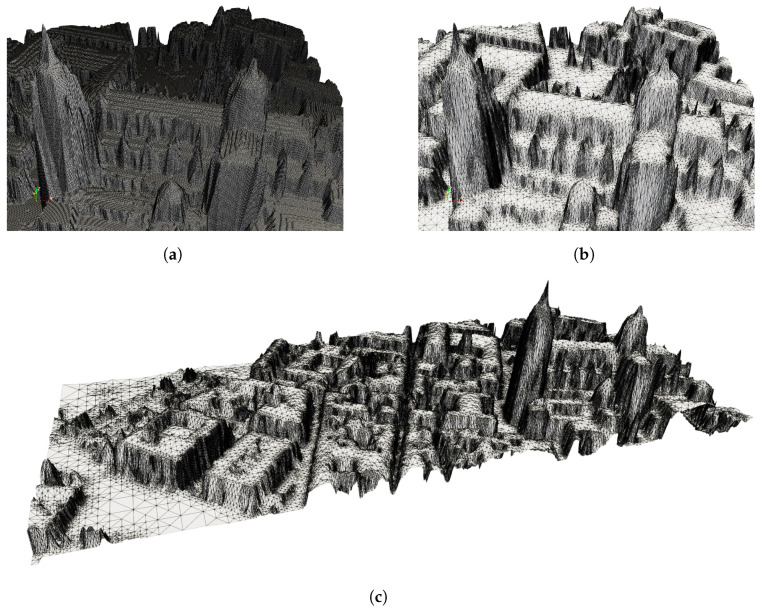
Adapted thin mesh (1 m) of the complete area (**c**), zoom of the fine adapted mesh over the Cathedral area (**b**), and detail of the uniform original mesh over the Cathedral area (**a**).

**Figure 12 sensors-24-01823-f012:**
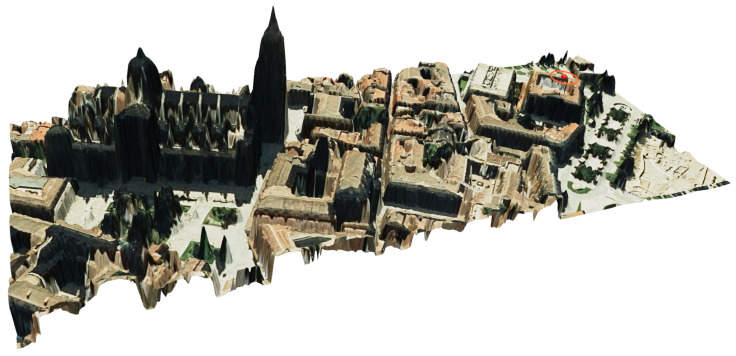
Three-dimensional reconstruction of the study area by simply projecting the orthophoto onto the fine adapted mesh. The pyranometers described in Section 2.5 are located at the red dot.

**Figure 13 sensors-24-01823-f013:**
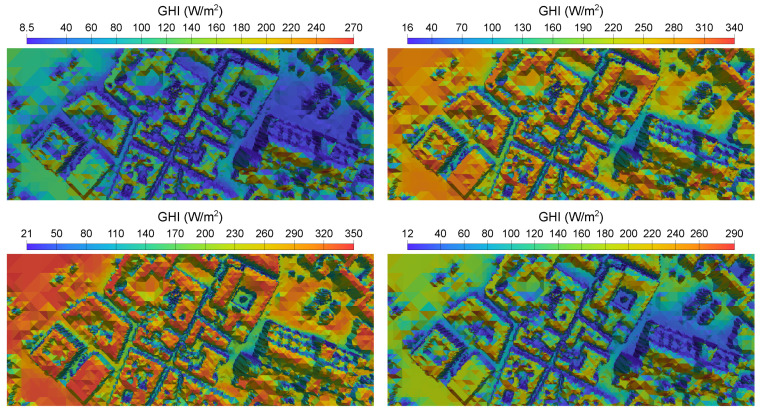
Mean GHI map for January, April, July and October, computed with MAPSol and coarse adapted mesh. Notice that the calculations assume a clear sky.

**Figure 14 sensors-24-01823-f014:**
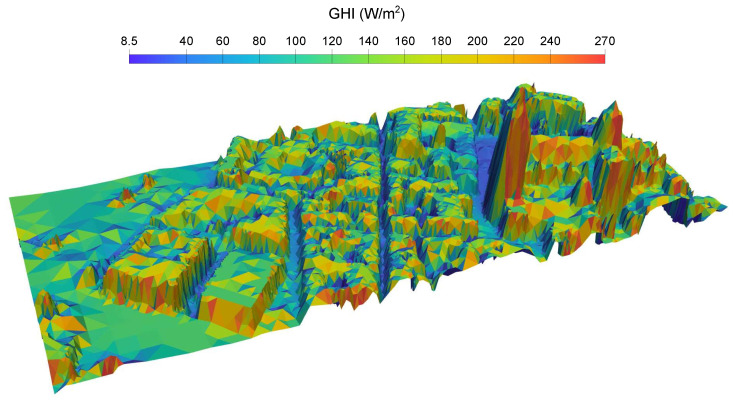
Annual mean GHI 3D map, computed with MAPSol and coarse adapted mesh. View from the south.

**Figure 15 sensors-24-01823-f015:**
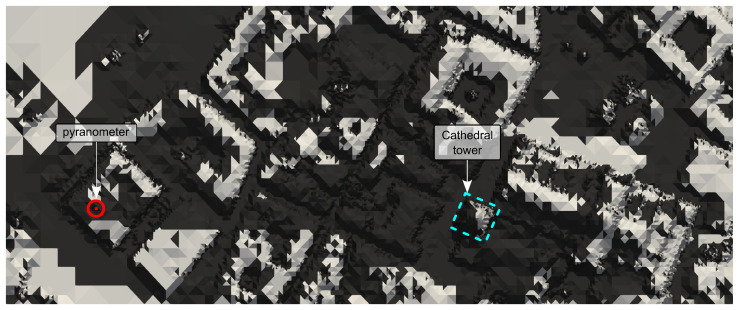
Shadow calculated with MAPSol on 4 September 2022 at 7.00 a.m. and coarse adapted mesh (Appendix A). The long shadow of the cathedral tower (blue dotted square) can be seen over the area where the pyranometers are located (red dot).

**Figure 16 sensors-24-01823-f016:**
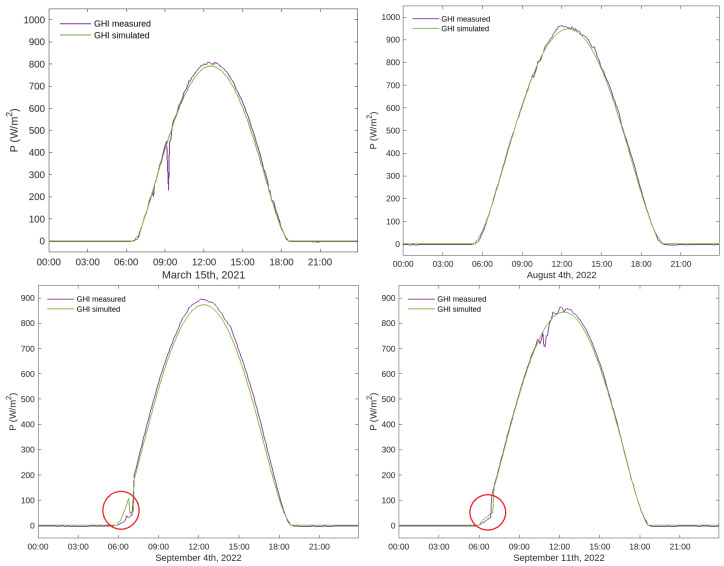
Curves of GHI measured (purple lines) and simulated (green lines) for the selected dates. The effect of the shadow of the cathedral tower at sunrise can be appreciated in the simulated and the measurement data (circled in red in bottom graphics). The accuracy of the irradiance fit calculated with the MAPSol model is very good, as can be seen in all of the graphs. The peak at 9 a.m. in the upper left graphic corresponds to a cloudy interval that affected the pyranometer readings, which cannot be simulated as the model assumes a clear sky.

**Table 1 sensors-24-01823-t001:** Main specifications of the LiDAR data of the first coverage offered by the IGN.

Feature	First Coverage
Minimum point density	$0.5\;{\mathrm{pt}/m}^{2}$
Years of flight	2009–2015
Geodetic reference system	ETRS89 zones 28, 29, 30 and 31 as appropriate
Altimetric reference system	Orthometric altitudes, reference geoid EGM08
RMSE Z	≤40 cm
Estimated planimetric accuracy	≤30 cm
File size	2×2km
File format	LAS 1.2 format 3

**Table 2 sensors-24-01823-t002:** Main features of the Kipp and Zonen SMP 10 pyranometers [50] used to perform the experimental measurements.

Feature	Value
Spectral range	285–2800 nm
Response time	(63%) < 0.7 s
Response time	(95%) < 2 s
Non-linearity	<0.2
Spectral selectivity	(350–1500 nm) < 3%
Field of view	180°

**Table 3 sensors-24-01823-t003:** Experimental and bibliographic values of accumulated energy by irradiance type in $\mathrm{kWh}\;m^{-2}{\mathrm{day}}^{-1}$ for each month of the year.

Source	AEMET [51]		Experimental Data		Relative Differences (%)	
**Month**	GHI	BHI	GHI	BHI	ΔGHI	ΔBHI
January	2.08	1.18	2.31	1.47	11.06	24.58
February	3.09	1.89	3.09	1.97	0.00	4.23
March	4.49	2.82	4.74	3.08	5.57	9.22
April	5.56	3.50	5.19	2.89	6.65	17.43
May	6.44	4.08	6.90	4.65	7.14	13.97
June	7.60	5.45	7.33	5.13	3.55	5.87
July	7.82	5.96	7.82	6.17	0.00	3.52
August	6.84	5.05	6.95	5.48	1.61	8.51
September	5.27	3.71	5.21	3.75	1.14	1.08
October	3.43	2.14	3.53	2.32	2.92	8.41
November	3.38	1.28	2.26	1.27	33.14	0.78
December	1.78	0.96	1.53	0.67	14.04	30.21

**Table 4 sensors-24-01823-t004:** Annual cumulative values (in kWh · m^−2^·yr^−1^) and average value per day (in kWh · m^−2^·day^−1^) of the energy received.

Source	AEMET [51]		Solargis [52]		Measured Records
Annual	Max.	Min.	Max.	Min.	
GHI	1708.2	1733.8	1680	1753	1733.65
BHI	1146.0	1182.6	−	−	1185.93
Daily	Max.	Min.	Max.	Min.	
GHI	4.68	4.75	4.6	4.8	4.75
BHI	3.14	3.24	−	−	3.25

**Table 5 sensors-24-01823-t005:** Summary of errors, in terms of MAE and MRSE and the corresponding normalized indicators NMAE and NMRSE, as well as the coefficient of determination $R^{2}$.

Date	MAE	NMAE	MRSE	NMRSE	$R^{2}$
15 March 2021	9.8632	1.2207	19.5744	0.0242	0.9959
4 August 2022	7.1667	0.7465	9.1455	0.0095	0.9994
4 September 2022	14.1427	1.5802	18.7152	0.0209	0.9971
11 September 2022	7.0888	0.8224	12.0603	0.0140	0.9986

## Data Availability

Publicly available datasets were analyzed in this study. These data can be found here: https://centrodedescargas.cnig.es/CentroDescargas (accessed on 31 January 2024).

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
