# Peer review of "Validation of a 3D Local-Scale Adaptive Solar Radiation Model by Using Pyranometer Measurements and a High-Resolution Digital Elevation Model"

_sensors, 2024, doi:10.3390/s24061823_

Round 1
Reviewer 1 Report
Comments and Suggestions for Authors
This paper try to create a model a 3D local-scale adaptive solar radiation by using pyranometer measurements and high-resolution DEM, This work will help us to provide an efficient protocol for studying solar resources, there are some questions for that paper,
1、Figure16 showed us the curves of GH I measured (purple lines) and simulated (green lines) for the selected dates,there is a spike at 9 am in the upleft figure, can the author explain how this spike generated between GHI measured and simulated?
2、In line 180, a shading level is allocated to shadowed triangles based on the count of warning points located within other triangles (refer to Figure 2), how to define the location of warning points for shading?
Author Response
Thank you for your comments which have allowed us to improve our work.
Regarding the first question, the peak at 9 a.m. in the figure on the left corresponds to a cloudy interval that affected the pyranometer readings, but cannot be simulated as the model assumes clear sky. We have added the corresponding explanation to the caption of figure 16, highlighted in red color.
Regarding the second question, we have improved the caption of Figure 2 as follows, again highlighted in red color in the document: Warning points for shading are equidistributed over each triangle (left graph) simply using the centers of the triangles obtained by refining with 4-T Rivara algorithm (right graph).
Reviewer 2 Report
Comments and Suggestions for Authors
Author Response
Thank you for your comments which have allowed us to improve our work.
Regarding the first question, we have changed the sentence:
“A simple system widely used incorporates a shadow ring, which is nothing more than a ring of an opaque material that casts a direct shadow on the receiver.”
By this new sentence (highlighted in red color) to improve the explanation about the shadow ring:
The standard procedure for determining diffuse radiation is by using a pyranometer (see Figure 4 and paragraphs below) shadowed by a shadowing ring, made of an opaque material of a calibrated width that avoids direct radiation on the receiver.
Regarding the second question, the key reason for selecting the 5-metre mesh is the large difference in operational cost, 68 times higher for the 1-metre mesh, while the improvement in the accuracy of the results is negligible. In fact, the only noticeable difference we have found is in a single data on 11 September, at 6.25 a.m., when the irradiance values are very low, so it is of no significance. This may be because the finer mesh picks up the geometry of the top of the cathedral tower somewhat better, but the effect is almost negligible.